# Case Report: A Detailed Phenotypic Description of Patients and Relatives with Combined Central Hypothyroidism and Growth Hormone Deficiency Carrying *IGSF1* Mutations

**DOI:** 10.3390/genes13040623

**Published:** 2022-03-30

**Authors:** Melitza S. M. Elizabeth, Anita Hokken-Koelega, Jenny A. Visser, Sjoerd D. Joustra, Laura C. G. de Graaff

**Affiliations:** 1Department of Internal Medicine—Endocrinology, Erasmus MC, University Medical Center Rotterdam, 3015 GD Rotterdam, The Netherlands; m.elizabeth@erasmusmc.nl (M.S.M.E.); j.visser@erasmusmc.nl (J.A.V.); 2Academic Center for Growth Disorders, Erasmus MC, University Medical Center Rotterdam, 3015 GD Rotterdam, The Netherlands; a.hokken@erasmusmc.nl; 3Department of Pediatrics, Subdivision Endocrinology, Erasmus MC, University Medical Center Rotterdam, 3015 GD Rotterdam, The Netherlands; 4Dutch Growth Research Foundation, 3016 AH Rotterdam, The Netherlands; 5Center for Genetics of Growth, Department of Pediatrics, Division of Pediatric Endocrinology, Willem-Alexander Children’s Hospital, Leiden University Medical Center, 2333 ZA Leiden, The Netherlands; sdjoustra@lumc.nl; 6Center for Adults with Rare Genetic Syndromes, Erasmus MC, University Medical Center Rotterdam, 3015 GD Rotterdam, The Netherlands

**Keywords:** *IGSF1*, pituitary hormones, hypothyroidism, growth hormone, genetic variation

## Abstract

In recent years, variants in immunoglobulin superfamily member 1 (*IGSF1*) have been associated with congenital hypopituitarism. Initially, *IGSF1* variants were only reported in patients with central hypothyroidism (CeH) and macroorchidism. Later on, *IGSF1* variants were also reported in patients with additional endocrinopathies, sometimes without macroorchidism. We studied *IGSF1* as a new candidate gene for patients with combined CeH and growth hormone deficiency (GHD). We screened 80 male and 14 female Dutch patients with combined CeH and GHD for variants in the extracellular region of *IGSF1,* and we report detailed biomedical and clinical data of index cases and relatives. We identified three variants in our patient cohort, of which two were novel variants of unknown significance (p.L570I and c.1765+37C>A). In conclusion, we screened 94 patients with CeH and GHD and found variants in *IGSF1* of which p.L570I could be of functional relevance. We provide detailed phenotypic data of two boys with the p.C947R variant and their large family. The remarkable phenotype of some of the relatives sheds new light on the phenotypic spectrum of *IGSF1* variants.

## 1. Introduction

The anterior pituitary secretes hormones that are crucial for the regulation of many physiological processes in the human body, such as metabolism, growth and development [1]. These hormones include growth hormone (GH), prolactin (PRL), thyroid-stimulating hormone (TSH), luteinizing hormone (LH), follicle-stimulating hormone (FSH), and adrenocorticotropic hormone (ACTH) [2]. Congenital hypopituitarism, the impaired production of pituitary hormones, is a rare disorder affecting 1:3000–1:10,000 live births [3,4]. 

Patients with congenital hypopituitarism show a wide variation in phenotype, depending on which hormones are deficient. Symptoms can range from neonatal hypoglycemia, prolonged neonatal jaundice, underdeveloped genitals, short stature, truncal obesity, delayed puberty, and, if the pituitary-adrenal axis is affected, it can sometimes even lead to death [4]. Understanding the etiology of congenital hypopituitarism is important for anticipation of clinical problems, for genetic counseling and to develop future treatment strategies for the disease.

Genetic variations in genes, such as *POU1F1*, *PROP1*, *HESX1*, *LHX3*, *LHX4*, *TRHR* and *TSHB*, have been associated with congenital hypopituitarism. In recent years, immunoglobulin superfamily member 1 (*IGSF1*), a new candidate gene located on the X-chromosome, has been the subject of extensive study. Initially, variants in *IGSF1* were reported to be associated with X-linked central hypothyroidism (CeH) in combination with adult macroorchidism in hemizygous male patients [5]. Subsequent large case series showed that affected males can also present with disharmonious pubertal development (normal or early timing of testicular growth, but delayed rise of testosterone), variable prolactin deficiency, increased body mass index, and decreased attentional control [5,6,7,8,9] and partial and transient growth hormone deficiency (GHD) in childhood (16% of cases). Sporadically, patients may show normal testicular volume [10,11]. All previously described pathogenic mutations in *IGSF1* led to loss of function in the protein. This phenomenon is called the IGSF1 deficiency syndrome, which is associated with variable endocrine and phenotypic characteristics [8,11,12,13]. 

Since 2003, The Dutch HYPOPIT study has investigated the genetic and non-genetic causes of congenital hypopituitarism. As *IGSF1* variants have been described in patients with GHD with normal testicular volume, we studied *IGSF1* as a possible candidate gene in a cohort of 94 patients with a combination of CeH and GHD. To give a complete overview, we describe all *IGSF1* variants found in this cohort and provide a detailed description of the phenotypes seen in the individuals carrying these variants. In addition, we provide the detailed phenotypic data of a large family including two boys with the pathogenic p.C947R variant. 

## 2. Materials and Methods

### 2.1. Cohort and Patient Selection

The Dutch HYPOPIT study is a multi-center study that includes six Dutch university hospitals and two regional hospitals. We used the Dutch National Registry of Growth Hormone Treatment to select all recombinant human growth hormone (rhGH)-treated children and adults that were registered between 1992 and 2003. From this group, we selected patients with combined GHD and CeH of unknown cause. To obtain an unbiased representation of the population of patients with combined GHD and CeH, each participating center had to include all eligible patients treated between 1992 and 2003, otherwise the center could not participate in the study. Approval was obtained from the Medical Ethics Committees of all participating hospitals. We obtained informed consent from all participating patients, as well as their parents if the patients were younger than 18 years. All patients were of Western European origin.

### 2.2. DNA Isolation

Genomic DNA of the patients was extracted from peripheral whole blood samples according to standard procedures. 

### 2.3. Mutation Screening 

All previously described intragenic pathogenic *IGSF1* mutations, except for (p.L569Ffs*16) are located in the extracellular domain of the IGSF1 protein (Figure 1), encoded by exons 10 to 17 [14]. Hence, we screened those exons of the *IGSF1* gene (NM_001170961.1) in 80 male and 14 female patients with combined CeH and GHD. Primer sequences and polymerase chain reaction (PCR) conditions were kindly provided by the Leiden University Medical Center. Exons 10-17 were amplified using PCR with TAQ polymerase (Qiagen, Venlo, The Netherlands). 

### 2.4. Sequencing

Sequencing was outsourced and performed by BaseClear B.V (Leiden, The Netherlands). For this purpose, 80 ng of DNA together with 25 pmol/µL of the primers was supplemented with sterile H_2_O. Forward and reverse primers were sequenced separately. The results were analyzed with Sequencher 4.2 [15]. The *IGSF1* sequence with accession number NM_001170961.1 was used as a reference sequence.

### 2.5. In Silico Analysis

We used in silico prediction tools, SIFT (https://sift.bii.a-star.edu.sg/, accessed on 21 November 2021) [16], PolyPhen (http://genetics.bwh.harvard.edu/pph2/, accessed on 21 November 2021) [17], PROVEAN (http://provean.jcvi.org/index.php, accessed on 21 November 2021) [18], FATHMM (http://fathmm.biocompute.org.uk, accessed on 21 November 2021) Mutation Taster V2.0 (https://www.mutationtaster.org/, accessed on 21 November 2021) [19] and VEP (https://www.ensembl.org/Tools/VEP, accessed on 21 November 2021) [20] to predict the pathogenicity of the novel variants.

### 2.6. Biomedical Examination 

We retrospectively collected patients’ clinical data and laboratory results which were performed as part of regular patient care. The same assays were used to carry out all measurements. 

## 3. Results

### 3.1. Screening of the Extracellular Region of IGSF1

We found a total of seven variants in *IGSF1* in our patients with combined CeH and GHD. Two novel variants (p.L570I and intronic variant c.1765+37C>A), a known pathogenic mutation (p.C947R) and four previously described benign variants. Genotypic details of these variants are shown in Table 1.

### 3.2. Novel Variants p.L570I and c.1765+37C>A

The two novel variants (p.L570I and intronic variant c.1765+37C>A) were identified in a female patient. She was born by breech delivery and showed neonatal hypoglycemia. Birth weight was 2440 g after 36 weeks of pregnancy. At presentation, she had a high-pitched voice, frontal bossing, strabismus and very poor vision. At the age of 3 years, her height was 81 cm (−5.1 SD), her bone age 1.0 years, and her weight 10 kg. IGF-1 was undetectable (<1.0 nmol/L) and IGF-BP3 was 1.3 mg/L (−3.8 SDS). 

GH stimulation with arginine and clonidine revealed a peak GH level of 5.5 and 6.2 mIU/L, respectively (reference men > 20 mIU/L). MRI revealed a very small sella turcica with a small anterior pituitary and an ectopic posterior pituitary. RhGH treatment was started at the age of 3 years. As her free thyroxine (FT4) was also low 7.6 pmol/L (reference range 9–22 pmol/L), levothyroxine treatment was started. Cortisol levels were normal and there was no diabetes insipidus. Her family history was unremarkable, and both her parents showed normal height.

We analyzed the variant with the VEP analysis tool from ensembl.org, accessed on 21 November 2021. In silico analysis predicted this variant to have a moderate modifying effect on the IGSF1 protein [Sift predicted this variant to be tolerated (*p* = 0.3); PolyPhen predicted this variant to be probably damaging (*p* = 0.9); LofToot predicted this variant to be possibly damaging (*p* = 0.38); CADDphred (score = 17.95); ClinPred predicted this variant to be possibly/probably damaging (see Table 2). Hence, we classified this as a variant of unknown significance (VUS), of which the exact relation with clinical phenotype is unknown. This variant is located 10 bp from a splice site. 

The intron variant c.1765+37C>A was also analyzed using VEP analysis tool from ensemble.org, accessed on 21 November 2021. Results for this analysis are shown in Table 2. In silico analysis with Mutation Taster V2 predicted this variant to be deleterious (*p*-value = 0.99, *p*-value close to 1 indicates a high ‘security’ of the prediction). This variant is located in a regulatory region 37 bp from a splice site, but its effect could not be predicted.

### 3.3. Pathogenic Mutation p.C947R

The p.C947R was found in two brothers, of which one (III-5) was previously mentioned briefly as individual H.III-3 in the publication by Sun et al., 2012 [5]. Here we describe detailed clinical, phenotypic and biochemical features of the two brothers and their large family who carried the pathogenic variant p.C947R. The family’s pedigree is shown in Figure 2 and a summary of clinical and biochemical features of the two boys are provided in Table 3 and Table 4. The clinical data of the affected and non-affected relatives is shown in Table 5. The most remarkable observations found in the two brothers and their relatives are described below.

### 3.4. Clinical Features of the Two Brothers with a p.C947R Variant

Index case III-5 is the third child of Dutch non-consanguineous parents. He was identified by the T4-TSH-TBG based Dutch neonatal screening program. His serum FT4 was low (10 pmol/L; reference range 13.5–19 pmol/L), with inappropriately low-normal serum TSH (2.6 mIU/L; ref 1.75–5 mIU/L). In a TRH stimulation test, he had a diminished increase of TSH in response to TRH, indicating a central cause of hypothyroidism. Thyroid ultrasound showed a severely hypoplastic thyroid gland [22]. Testicular volume was 6 mL at the age of eleven years (1.5 SDS [21]). He had an elevated serum FSH of 13.2 mIU/L (ref 4–5 mIU/L), low LH and testosterone, low SHBG, elevated AMH (18.3 µg/L; reference value 9 µg/L) and high inhibin-B (424 pg/L; reference range 131–227 pg/mL) [23]. He showed prepubertal responses to GnRH throughout infancy. His α-subunit concentration was normal. At the age of 12 years, his testicular volume was 10 mL (1.3 SDS). Testicular volume was last measured at 15 years old (30 mL, 1.8 SDS) [21]. Apart from CeH and borderline macroorchidism, he had short stature and partial GHD (GH peak 7.7 mIU/L at arginine test and 14.3 mIU/L at clonidine test). His pituitary MRI was normal. He was treated with rhGH and levothyroxine. 

His brother, III-4, was not detected by neonatal screening but was diagnosed with and treated for CeH at 7 years of age. His serum FT4 was low (8 pmol/L; reference range 13.5–19 pmol/L), with inappropriately normal serum TSH (3.6 mIU/L; reference range 1.75–5 mIU/L). TSH response to TRH during a stimulation test was at the lower limit of normal (Figure 3). Thyroid ultrasound revealed hypoplastic thyroid tissue (0.28 mL; reference value 2–5 mL), without nodules or cysts [24]. MRI showed no pituitary abnormalities. Like his brother, he had above-average testicular growth. He had prepubertal levels of FSH, LH, testosterone and responses to GnRH. At 12.5 years, his testicular volume was 12 mL (1.2 SDS), reaching > 30 mL at 14.5 years (>2.0 SDS). His α-subunit levels were normal, but inhibin-B and AMH were strongly elevated. He did not have GHD. 

### 3.5. Genotype and Phenotype Description of the Relatives

We collected the phenotypic data from the brother, parents and uncle of the two affected boys. The older brother of the two boys, III-3 with wild-type (WT) genotype, was healthy and had a normal physical appearance. The father of the brothers, II-5 (WT), and the mother (Carrier) were healthy, with a normal appearance. They did not report any symptoms of hypothyroidism. The mother’s pregnancies were unremarkable, with no miscarriages, but she had hyperemesis gravidarum in all three pregnancies. She had a normal age of breast development, although the exact date of menarche was unknown. She voluntarily chose not to give breastfeeding, not because lactation failed. However, she experienced an early menopause at the age of 44 years. She has two brothers and one sister. 

The mother’s youngest brother II-9 (Affected), uncle of the two boys and hemizygous for the same p.C947R mutation, had a remarkable phenotype. As a child, he had delayed psychomotor development, growth retardation and slow speech. Hypothyroidism (without goiter) was diagnosed, for which he received levothyroxine. After the start of LT4, his school results improved drastically. His ACTH test was normal, and X-sella showed a normal-sized sella turcica. Abdominal ultrasound was unremarkable. His adult height is 190 cm without rhGH treatment. As a child, he had undescended testes and at the age of 16 years he was still prepubertal. His testosterone levels were low (0.3 ug %, ref 0.4–1.2 ug %). After he received hCG treatment, his testes descended and, like his nephews, he reported above-average testicular growth. Although not measured by ultrasound or orchidometer, his self-reported final testicular volume was large. He reported a remarkable decrease in testicular volume over the years. At the age of 50, he consulted the GP due to fatigue, after which primary hypogonadism was diagnosed. He had two sons (III-11 and III-12), without endocrine anomalies.

The phenotypes of the mother’s other brother and sister were unremarkable. The maternal grandmother of the two boys (I-9) was healthy, with a normal appearance. Like all her sisters, she entered puberty late (aged 16 year). She had no fertility issues and entered menopause at the age of 46 y. She had no symptoms of hypo- or hypothyroidism and no goiter.

Apart from the two boys and uncle II-9, there were no thyroid problems in the family. However, a remarkable finding in this pedigree was that according to the grandmother of the boys, all her sisters (I-9,10,11 and 12) had delayed puberty. 

## 4. Discussion

Although the classical IGSF1 deficiency syndrome includes macro-orchidism and central isolated hypothyroidism, *IGSF1* variants have also been described in patients with GHD and normal testicular volume. Therefore, we studied *IGSF1* as a possible candidate gene in a unique cohort of 94 Dutch patients with the combination of CeH and GHD. 

To give a complete overview, we describe all *IGSF-1* variants found in this cohort and provide a detailed phenotypic description of the individuals carrying these variants. In addition, we provide detailed phenotypic data of a large family including two boys with the pathogenic p.C947R variant. We describe a novel missense variant (p.L570I) combined with a novel intronic variant c.1765+37C>A of unknown significance and a pathogenic mutation p.C947R in a family with a remarkable phenotype. 

Loss-of-function variants in *IGSF1* result in CeH, either isolated or in combination with additional pituitary hormone deficiencies [9]. *IGSF1* is a highly polymorphic gene and, to date, 136 genetic variants have been reported. An overview of variants associated with endocrinopathies is shown in Figure 1. Nearly all intragenic variants described to date are present in the C-terminal domain of *IGSF1,* and therefore we focused our screening on that part of the protein. Even though *IGSF1* has been the subject of extensive study, the exact function of *IGSF1* is still not determined [14]. 

Apart from the known variants and SNPs we found, there were several interesting findings. In one female patient with CeH and GHD, we found a novel missense variant (p.L570I) combined with the novel intronic variant c.1765+37C>A. The novel variant, p.L570I, is located in the hydrophilic linker domain of *IGSF1*. Two other variants in the same domain, p.A532V and p.L569Ffs*16, have been associated with endocrinopathies. Furthermore, p.L569Ffs*16 has been shown to impair plasma membrane trafficking of the protein [25]. In silico analysis of this variant was inconclusive; the fact that p.L570I is located next to L569Ffs*16 suggests that p.L570I could possibly affect plasma membrane trafficking.

The male relatives who were hemizygous for the p.C947R variant showed a variable and remarkable phenotype. The index case and his brother presented with CeH, near-macroorchidism, and elevated inhibin-B and AMH levels. Their uncle, who had the same mutation, had hypothyroidism, delayed psychomotor development, growth retardation and slow speech as a child. Although not measured by ultrasound or orchidometer, his self-reported final testicular volume was very large. He reported a remarkable decrease in testicular volume over the years and was diagnosed with primary hypogonadism at the age of 50.

The increased AMH/InhB in the two brothers is remarkable, since AMH and inhibin levels are reported to be generally normal in patients carrying *IGSF-1* mutations [6]. This might indicate an increased number or functioning of Sertoli cells, and therefore possibly explain the morphologic characteristics of the large testes. 

There are some limitations to this study. As we describe a historical cohort, we were unable to perform additional genetic tests in some relatives and we were also unable to perform functional testing for some interesting variants. However, the detailed phenotypic data that we provide about the large family, including two boys with the pathogenic p.C947R variant, and the fact that we provide a complete overview of all *IGSF-1* variants found in this unique cohort of patients with GHD and CeH, adds relevant information to the existing literature

## 5. Conclusions

We screened 94 patients with CeH and GHD and found several variants, of which p.L570I, based on previously reported data, could be of functional impact. We provide detailed phenotypic data of two boys with the p.C947R variant and their large family. The remarkable phenotype of some of the relatives sheds new light on the phenotypic spectrum of IGSF1 variants. 

## Figures and Tables

**Figure 1 genes-13-00623-f001:**
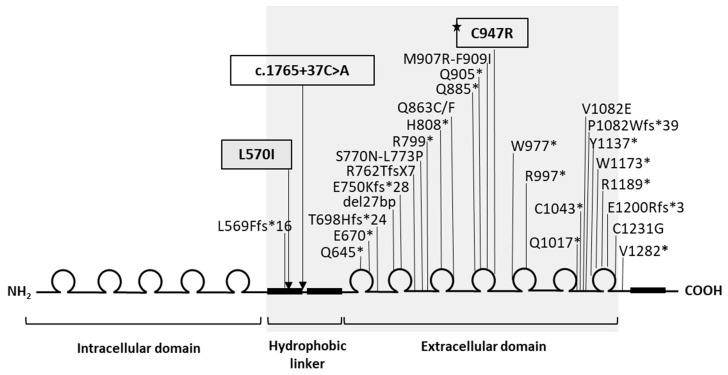
Schematic representation of IGSF1 protein domain structure and the relative locations of the mutations identified in this study and previously reported pathogenic mutations. The two novel variants (p.L570I and c.1765+37C>A) are framed in black. The pathogenic variant is labeled with (★). Region screened in this study (grey).

**Figure 2 genes-13-00623-f002:**
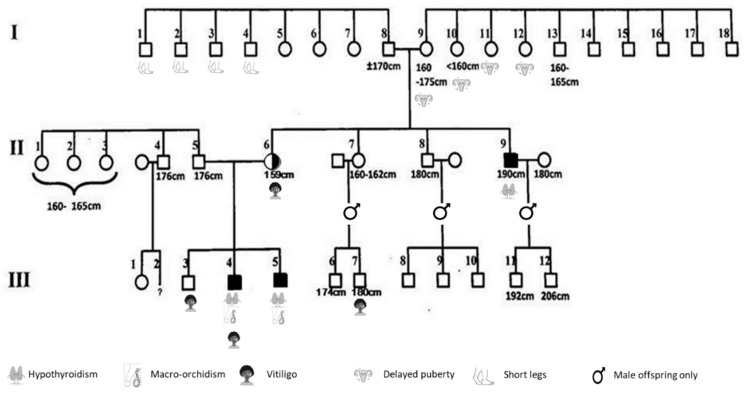
Family pedigree of the two brothers with the p.C947R variant showing affected and carrier family members. Numbers I, II and III indicate the first, second and third generation. Closed figures = hemizygous p.C947R variant; half open figures = carrier.

**Figure 3 genes-13-00623-f003:**
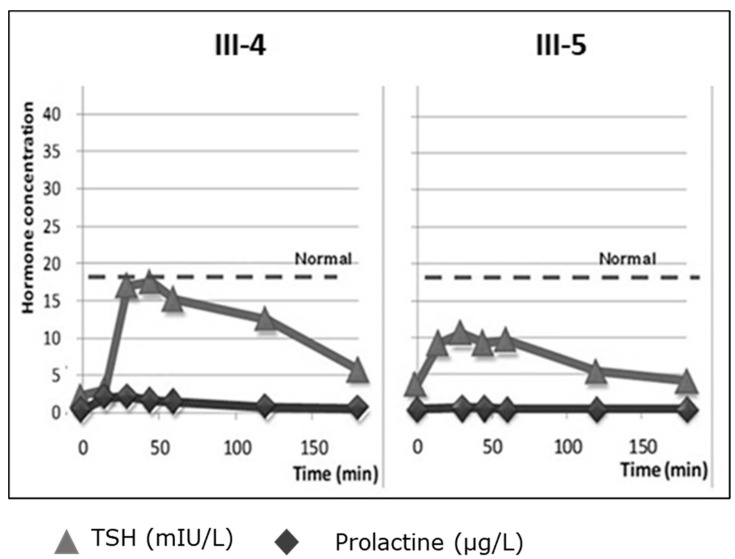
TRH stimulation tests results of patient III-4 and III5 with the p.C947R variant. *Y*-axis shows TSH (mIU/L) (in grey) and prolactine (ug/L) hormone concentrations (in black).

**Table 1 genes-13-00623-t001:** Overview of variants found in the extracellular regions of *ISGF1* in 94 Dutch patients with a combination of CeH and growth hormone deficiency. VUS = Variance of uncertain significance; HL = Hydrophobic linker domain.

cDNA Seq	Protein Seq	Type of Variant	Ig Loop	Reference	Variation Classification	ACMG Classification	GNOMAd MAF European Population
c.1709C>A	p.L570I	Missense	HL	Novel variant	VUS	US (ii)	Variant not found in online data sets
c.1765+37C>A		Intron		Novel variant	VUS	US (i)	Variant not found in online data sets
c.2839 T>C	p.C947A	Missense	9	[5]	pathogenic	PS3	

**Table 2 genes-13-00623-t002:** Summary of in silico analysis by VEP analysis.

Variant ID	Sift	PolyPhen	LofTool	CADDphred	ClinPred	GERP++NR	MexEntScan
p.L570I	0.3	0.942	0.379	17.95	D	5.04	-
c.1765+37C>A	-	-	-	-	-	-	-

**Table 3 genes-13-00623-t003:** Longitudinal data on testicular volume, FSH, LH and testosterone levels of the two brothers with a p.C947R variant. Testicular volume according to Tanner, standard deviation score is indicated between brackets [21].

**Patient III-4**									
**Age**	**Unit**	**7** **Years**	**7.5** **Years**	**10** **Years**	**11** **Years**	**13** **Years**	**14** **Years**	**15** **Years**	
LT4 dose	µg/kg/day	0	3	3	2	1.5	NA	NA	
TSH	mIU/L	3.6	<0.005	0.002	<0.001	<0.001	<0.001	<0.001	
FSH	U/L							6.1	
LH	U/L							1.3	
Testosteron	nmol/L				0.3	0.3	2.1	7	
Testicular volume	mL	2			3	12 (0.7 SDS)	18 (0.6 SDS)	30 (1.8 SDS)	
**Patient III-5**									
**Age**	**Unit**	**6** **Weeks**	**3** **Years**	**10** **Years**	**11** **Years**	**12** **Years**	**13** **Years**	**13.5** **Years**	**14.5** **Years**
LT4 DOSE	µg/kg/day	0	3	2	NA	NA	NA	NA	2
TSH	mIU/L	6	<0.05	0.29					<0.001
FSH	U/L	4.6				0.3			
LH	U/L	0.4				4.6			
Testosteron	nmol/L					0.2	2.2	5	5.2
Testicular volume	mL	2		3 (0.6 SDS)	6 (1.5 SDS)	8 (1.0 SDS)	16 (1.2 SDS)	23 (1.6 SDS)	>30 (>2.0 SDS)

**Table 4 genes-13-00623-t004:** Biochemical data of the affected and non-affected relatives.

		**III-4**			**III-5**			**III-3**	**III-6**	**III-7**	**Reference Range**			
	Unit	A	B	C	A	B	C							
Genotype		Affected			Affected			WT	WT	WT				
Sex		Male			Male			Male	Male	Male	Male		Female	
TSH	mIU/L	0.002 ↓	45.6 ↑	30.6 ↑	0.001 ↓	37.2 ↑	26.0 ↑	1.77	0.77	1.89	0.4	4.3	0.4	4.3
T4	nmol/L	73	27.1 ↓	27.5 ↓	86	10.9 ↓	10.5 ↓	72.5	90	84.00	0.4	4.3	0.4	4.3
FT4	pmol/L	14.6	3.9 ↓	4.1 ↓	16.2	1.0 ↓	1.1 ↓	13	18.3	16.30	9	23	9	23
T3	nmol/L	1.49	1.23 ↓	1.32 ↓	1.77	0.66 ↓	0.68 ↓	1.87	1.98	2.19	11	25	11	25
FSH	U/L	4	5.6	5.3	10.0 ↑	13.2 ↑	10.0 ↑	3.27	2.12	1.84	1.4	2.5	1.4	2.5
LH	U/L	0.7 ↓	3.2	2.6	1.5	2.6	1.2 ↓	5.17	3.84	1.23	2	7	1	15
SHBG	nmol/L	4.6 ↓	2.7 ↓	2.9 ↓	12	8.2 ↓	7.0 ↓	14.1	18.8	27.60	1.5	8	15	90
Testosterone	nmol/L	5.8 ↓	12	8.3 ↓	9.9 ↓	17.7	13.2	15	24	16.50	10	70	20	120
Estradiol	pmol/L	58	138	106	10 ↓	130	75	176			10	30	0.5	3
Inhibin B	ng/L	344	301	405 ↑	351	424 ↑	348	196	317	360	150	400	<10	
AMH	µg/L	17.5 ↑	18.5 ↑	18.0 ↑	15.9 ↑	18.3 ↑	17.7↑	15.06			5.1	9.1	2	14
		**II-5**		**II-6**	**II-7**		**II-8**	**II-9**		**I-8**	**Reference Range**			
Genotype		WT		Affected	WT		WT	Affected		WT				
Sex		Male		Female	Female		Male	Male		Male	Male		Female	
TSH	mIU/L	3.69		0.82	1.31		0.14 ↓	0.28 ↓		0.6	0.4	4.3	0.4	4.3
T4	nmol/L	81.1		98	85		93	73		114	0.4	4.3	0.4	4.3
FT4	pmol/L	14.3		15.5	16.8		19.4	15.7		22.3	58	128	58	128
T3	nmol/L	2.36		1.81	2.02		1.97	1.5		2.04	11	25	11	25
FSH	U/L	4.54		44.9	69.5		2.56	18.7 ↑		58.8	1.4	2.5	1.4	2.5
LH	U/L	3.3		12 ↓	34.5		4.64	4.1		15.6	2	7	1	15
SHBG	nmol/L	24		32.9	61.6		22.6	19.4		67.1	1.5	8	15	90
Testosterone	nmol/L	12.4		0.8	0.8		14.8	9.0 ↓		0.4 ↓	10	70	20	120
Estradiol	pmol/L	77		46							10	30	0.5	3
Inhibin B	ng/L	188		<10 ↓	< 10 ↓		196	113 ↓		<10 ↓	150	400	<10	
AMH	µg/L	12		<0.1				2.1 ↓			5.1	9.1	2	14

The normal range for males and females is shown in the column on the right. ↓ value below the lower limit of normal; ↑ = value above the upper limit of normal.

**Table 5 genes-13-00623-t005:** Overview of clinical features of the two brothers with a p.C947R variant and their first- and second-degree relatives.

	III-4	III-5	III-3	III-6	III-7	III-11	III-12	II-5	II-6	II-7	II-8	II-9	I-2	I-8	I-9	I-3	I-5
Genotype	Affected	WT	Affected	WT	WT	WT	WT	WT	Affected	WT	WT	Affected	WT	WT	WT	WT	WT
Gender	Male	Male	Male	Male	Male	Male	Male	Male	Female	Female	Male	Male		Male	Female		
Adult height (CM)	NA	normal with rhGH treatment	NA	174	180.6	192	206	176	159	162.5	180.6	186.4		170	153.5		
Weight				73	80			86	88	67.2	83				62.4		
Sitting height				91	92.9			94.8	85.2	84.8	95	97					
Head size	UNR	UNR	UNR	UNR	UNR	UNR	UNR	UNR	Normal (53 cm)	Normal (56 cm)	UNR	UNR		UNR	UNR		
Thyroid size	HP	HP	UNR	UNR	UNR	UNR	UNR	UNR	Normal	Normal	Normal	Normal	UNR	UNR	Normal		
Thyroid function	CeH	CeH	Normal	CET	CET	Normal	Normal	CET	CET	CET	CET	CET		CET	CET		
Testes	Large-normal	Enlarged	Normal	Normal	Normal	Normal	Normal	Normal	NA	NA	Normal	Cryptorchism until Pregnyl		NA	NA		
Pituitary imaging	Normal	Normal	NA	NA	NA	NA	NA	NA	NA	NA	NA	NA		NA	NA		
Psychomotor Development	Normal	Normal	Normal	Slightly delayed, adult: Normal	UNR	UNR	UNR	UNR	UNR	UNR	UNR	slow (until Pregnyl/ LT4)	UNR	UNR	UNR	UNR	
Onset of puberty *	Normal	Normal	Normal	Normal	Normal	Normal	Normal	Normal	Delayed (16 y)	Normal	Normal	Delayed (Pregnyl)		NA	Delayed (16 y)		
Fertility	NYK	NYK	NYK	NYK	NYK	NYK	NYK	Normal	Normal	Normal	Normal	Normal		Normal	Normal		
Menopause	NA	NA	NA	NA	NA	NA	NA	NA	44	50	NA	NA	NA	NA	46	NA	
Other						Benign cerebellar tumor		Only male offspring	Only male offspring	Only male ffspring	Only male offspring	Only male offspring	Shortlegs			Short legs	Short legs

* Onset of puberty as estimated by patients and relatives based on physical appearance (not based on testicular volume or serum hormone levels). NA = not assessed; UNR = unremarkable; CET = clinically euthyroid; NYK = not yet known; HP = hypoplastic.

## Data Availability

The data presented in this study are available in this article.

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
