# Peer review of "Case Report: A Detailed Phenotypic Description of Patients and Relatives with Combined Central Hypothyroidism and Growth Hormone Deficiency Carrying IGSF1 Mutations"

_genes, 2022, doi:10.3390/genes13040623_

Round 1

Reviewer 1 Report

Review Manuscript ID: genes-1502128

IGSF1 variants in a cohort of patients with combined central hypothyroidism and growth hormone deficiency: detailed phenotypic description of patients and relatives
Melitza Elizabeth * , Anita Hokken-Koelega , Jenny Visser , Sjoerd Joustra , Laura de Graaff

Summary:
In this article 94 patients with central hypothyroidism (CeH) and growth hormone deficiency (GHD) were screened for variants in the extracellular domain of IGSF1. The authors look for variants in IGSF1, but do not find new pathogenic variants. They do find two VUS, but do not perform gene expression studies. Variants classified as benign or likely benign should not be reported. The added value and novelty of the article is not clear. The authors describe a case report of two siblings with a known pathogenic variant, this might be interesting, but then the article is rather a case report.

General concept comments:
IGSF1 is known to be responsible for hypothyroidism and testicular enlargement. The authors identified 7 variants in IGSF1 patients with combined CeH and GHD.
However:
- only one variant is pathogenic (already described)

- four variants are predictes as being benign, and are present in a high proportion of the European population (pV985A and pN604T, pY857= and pK721=)

- the novel variant c.1765+37 is located at 37 bp from a splice site and it is predicted not to affect splicing, please add also for this variant the pathogenicity prediction and ACMG classification

- the novel variant p.L561I seems also to have a tolerant prediction

Specific comments:
- Variants classified as benign or likely benign should not be reported.

- Further gene expression studies are essential for the VUS variants or another case with same mutation could also increase the possibility of pathogenicity (a case recruited from another center, or recruited through genematcher)

Author Response

Dear Editor, dear reviewers,

We would like to thank you for your very helpful comments on our manuscript entitled ‘IGSF1 variants in a cohort of patients with combined central hypothyroidism and growth hormone deficiency: detailed phenotypic description of patients and relatives’.

As this is the second revision, we only address the new comments and the new changes we made to the manuscript. We are grateful for the comments and we strongly believe that the suggested adaptations have helped to significantly improve the manuscript.

Please find our response below.

Reviewer 1

Comment 1: The novel variant c.1765+37 is located at 37 bp from a splice site and it is predicted not to affect splicing, please add also for this variant the pathogenicity prediction and ACMG classification.

Thank for your comment. We added the ACMG classification of this variant to the table.

Comment 3: The novel variant p.L570I seems also to have a tolerant prediction.

Thank you for your comment.  Additionally, we performed a VEP analysis via Ensambl.org. This analysis reported this variant to have a moderate modifying effect on the IGSF1 protein. We also added a table with a summary of the results from the VEP

Comment 4: Variants classified as benign or likely benign should not be reported.

Thank for your comment. We adjusted the manuscript and removed all previously reported benign variants from the manuscript and hereby we only report the pathogenic and novel variants found in our patient cohort.

Reviewer 2 Report

This paper presents valuable data and deserves publication.

Little bit disappointed that the authors limited the search to exons 10-17, instead of entire coding sequence and some flanking intronic regions

Figure 1. Authors examined exons 10-17. In the picture, the variants are shown so the reader assumes that the numbers along sequence may represent exons which is misleading in my opinion.

Please clarify the meaning of these numbers along protein sequence. In general, instead of adopting this figure I would encourage authors to prepare a new better one.  I think it is pretty valuable for the paper.

Table 1

First column – Sea data: use regular “c” instead of “C” in capital

L66: IGSF-1 should be IGSF1

L88: L569Ffs*16 should be p. L569Ffs*16

L108: Mutation Tasters ???? As far as I know only Mutation Taster v2 is available now. Please correct

L129 and 132: no spacing in variant description p.Y857= instead of p.Y857 =

Results

Novel variants p.L561I and c.1765+37C>A.

In order to find more convincing data, I will strongly encourage to use additional algorithms for splice site variants than relay only on Mutation taster splice prediction.  i.e. MaxEntScan (available via VEP in Ensembl). Please look also for evolutionary conservativeness  of those postions.

Author Response

Dear Editor, dear reviewers,

We would like to thank you for your very helpful comments on our manuscript entitled ‘IGSF1 variants in a cohort of patients with combined central hypothyroidism and growth hormone deficiency: detailed phenotypic description of patients and relatives’.

As this is the second revision, we only address the new comments and the new changes we made to the manuscript. We are grateful for the comments and we strongly believe that the suggested adaptations have helped to significantly improve the manuscript.

Please find our response below.

Reviewer 2:

Comment 1: Figure 1. Authors examined exons 10-17. In the picture, the variants are shown so the reader assumes that the numbers along sequence may represent exons which is misleading in my opinion.

Please clarify the meaning of these numbers along protein sequence. In general, instead of adopting this figure I would encourage authors to prepare a new better one.  I think it is pretty valuable for the paper.

Thank you for your very useful comment and suggestion. We made a new figure that is  better represents our data.

Comment 2: Table 1 First column – Sea data: use regular “c” instead of “C” in capital.

Thank you for your comment. We changed the format of the table, which now allows a ‘c’ instead of ‘C’.

Comment 3: L66: IGSF-1 should be IGSF1

Thank you for your attentive comment. We changed  IGSF-1 to IGSF1

Comment 4 : L88: L569Ffs*16 should be p. L569Ffs*16

Thank you for your comment. We changed  L569Ffs*16 to p. L569Ffs*16

Comment 5: L108: Mutation Tasters ???? As far as I know only Mutation Taster v2 is available now. Please correct

Thank you for your comment. We changed and added the version of mutation taster V2.

Comment 6: L129 and 132: no spacing in variant description p.Y857= instead of p.Y857 =

Thank you for your comment. This text was removed from the manuscript.

Comment 7:  Novel variants p.L570I and c.1765+37C>A. In order to find more convincing data, I will strongly encourage to use additional algorithms for splice site variants than relay only on Mutation taster splice prediction.  i.e. MaxEntScan (available via VEP in Ensembl). Please look also for evolutionary conservativeness  of those positions.

Thank you for your comment. Additionally, we performed a VEP analysis via the ensemble database. This analysis reported this variant to have a moderate modifying effect on the IGSF1 protein. We also added a table with a summary of the results from the VEP analysis. CAD and GERP++ scores were also added to the table. 

Once again, we thank you for your helpful comments!

Reviewer 3 Report

Very well written study. Congartulations.
